# Biodegradation of Polylactic Acid-Based Bio Composites Reinforced with Chitosan and Essential Oils as Anti-Microbial Material for Food Packaging

**DOI:** 10.3390/polym13224019

**Published:** 2021-11-20

**Authors:** Teuku Rihayat, Agung Efriyo Hadi, Nurhanifa Aidy, Aida Safitri, Januar Parlaungan Siregar, Tezara Cionita, Agustinus Purna Irawan, Mohammad Hazim Mohamad Hamdan, Deni Fajar Fitriyana

**Affiliations:** 1Department of Chemical Engineering, Politeknik Negeri Lhokseumawe, Lhokseumawe 24301, Indonesia; 2Mechanical Engineering Department, Faculty of Engineering, Universitas Malahayati, Bandar Lampung 35153, Indonesia; efriyo@malahayati.ac.id; 3Department of Renewable Energy Engineering, Universitas Malikussaleh, Muara Batu 24355, Indonesia; nurhanifaaidy@gmail.com; 4Department of Chemical Engineering, Faculty of Engineering, Universitas Sumatera Utara, Kota Medan 20222, Indonesia; aidasafitri853@gmail.com; 5College of Engineering, Universiti Malaysia Pahang, Gambang 26300, Malaysia; januar@ump.edu.my; 6Department of Mechanical Engineering, Faculty of Engineering and Quantity Surveying, INTI International University, Seremban 71800, Malaysia; tezara.cionita@newinti.edu.my; 7Faculty of Engineering, Universitas Tarumanagara, Jakarta Barat 11440, Indonesia; agustinus@untar.ac.id; 8Faculty of Engineering and Computing, First City University College, Petaling Jaya 47600, Malaysia; hazim.hamdan@firstcity.edu.my; 9Department of Mechanical Engineering, Universitas Negeri Semarang, Semarang 50229, Indonesia; deniifa89@mail.unnes.ac.id

**Keywords:** polylactic acid, turmeric essential oil, chitosan, TGA, FT-IR, antimicrobial and antioxidant properties, *S. aureus*, *E. coli*

## Abstract

This study aims to produce and investigate the potential of biodegradable Polylactic Acid (PLA)-based composites mixed with chitosan and Turmeric Essential Oil (TEO) as an anti-microbial biomaterial. PLA has good barrier properties for moisture, so it is suitable for use as a raw material for making packaging and is included in the GRAS (Generally Recognized As Safe). Chitosan is a non-toxic and antibacterial cationic polysaccharide that needs to be improved in its ability to fight microbes. TEO must be added to increase antibacterial properties due to a large number of hydroxyl (-OH) and carbonyl functional groups. The samples were prepared in three different variations: 2 g of chitosan, 0 mL TEO and 0 mL glycerol (Biofilm 1), 3 g of chitosan, 0.3 mL TEO and 0.5 mL of glycerol (Biofilm 2), and 4 g of chitosan, 0.3 of TEO and 0.5 mL of glycerol (Biofilm 3). The final product was characterized by its functional group through Fourier transform infrared (FTIR); the functional groups contained by the addition of TEO are C-H, C=O, O-H, and N-H with the extraction method, and as indicated by the emergence of a wide band at 3503 cm^−1^, turmeric essential oil interacts with the polymer matrix by creating intermolecular hydrogen bonds between their terminal hydroxyl group and the carbonyl groups of the ester moieties of both PLA and Chitosan. Thermogravimetric analysis (TGA) of PLA as biofilms, the maximum temperature of a biofilm was observed at 315.74 °C in the variation of 4 g chitosan, 0.3 mL TEO, and 0.5 mL glycerol (Biofilm 3). Morphological conditions analyzed under scanning electron microscopy (SEM) showed that the addition of TEO inside the chitosan interlayer bound chitosan molecules to produce solid particles. Chitosan and TEO showed increased anti-bacterial activity in the anti-microbial test. Furthermore, after 12 days of exposure to open areas, the biofilms generated were able to resist *S. aureus* and *E. coli* bacteria.

## 1. Introduction

Plastic manufacturers are facing a new challenge in the food industry as a result of the growing demand for high-quality food that is free of preservatives. This is because there is a greater emphasis on developing conservation products and anti-microbials for renewable and environmentally friendly products. Furthermore, manufacturers’ use of natural packaging materials (biofilms) as an alternative extends the life of a product [1,2]. Food packaging protects food products from external factors such as microorganisms, moisture, and ultraviolet (UV) light, thereby extending their shelf life. Recently, biopolymer-based food packaging has attracted a lot of attention in terms of environmental issues due to the fact that plastics constitute a considerable portion of buried garbage in the natural ecosystem and are regarded as emerging contaminants with significant environmental effects due to their high concentration, extensive dispersion, and non-biodegradability. This is considerable evidence of aquatic plastic pollution, including plastic islands and microplastics [3]. Conventional plastics are a vital commodity owing to their particular lightweight, thermostable, crystalline, and easy-to-mold properties, resulting in a wide range of items that combine comfort and quality in our lives. Increased usage of plastic materials in the home and industrial sectors has outpaced worldwide production by up to 400 Mt/year, raising severe issues about disposal, environmental pollution, toxicity to the ecosystem, and human health [4].

The selection of anti-microbial agents used as packaging materials must adhere to established regulations, particularly in terms of toxicological effects [5]. Chitosan, a biodegradable polymer, has been widely used in the production of edible films because it is non-toxic and environmentally friendly. Chitosan also has excellent film-forming ability, massive antimicrobial activity, selective permeability to gases (CO_2_ and O_2_), and it is compatible with other substances, such as vitamins and minerals [6]. Furthermore, the use of antimicrobial agents in biodegradable food packaging systems to prevent microbial growth on food surfaces has been the focus of recent research. Several previous researchers [7,8] have investigated the use of chitosan as an additional material in packaging manufacturing. The use of chitosan in food packaging represents a renewal based on a biodegradable food packaging concept in which the spread of packaging can reduce microorganisms in foods and prevent their development [9]. Chitosan-derived natural food packaging (biofilm) can improve melons’ quality and extend their shelf life. It can also inhibit the growth of microbes, bacteria, and fungi.

Chitosan is an antimicrobial agent that contains polysaccharides and amino groups. To enhance chitosan’s antimicrobial activity, several other chemicals were added, including a biopolymer, polysaccharides, lipids, or a mixture of these, as well as fatty acids and essential oils [10,11]. According to previous research, the inclusion of turmeric (essential oils) in film materials may protect L. inocula from UV light contamination. This is due to the fact that UV light can be rendered inactive. When compared to the three-hour exposure period for pure chitosan, adding turmeric extract to chitosan can significantly boost anti-microbial action and reduce the number of *Staphylococcus aureus* and *Salmonella bacteria* [12]. Formalized paraphrase Curcumin is also one of the naturally occurring hydrophobic polyphenols and is an ayurvedic treatment in the Indian traditional medicine system as well as in many Asian countries. Curcumin can be found in turmeric and it is well known for its antimicrobial, antioxidant, anticancer, and anti-inflammatory properties [13].

Renewable aliphatic polyesters of homopolymers and copolymer-type polyhydroxylic acid composed of PLA, Poly Glycolic Acid (PGA), and Poly e-Caprolactone (PCL) are the most promising materials for biofilm applications. PLA has received special attention as a replacement for traditional petroleum-based plastics. Based on a study [14], biodegradable films with antioxidant activity for active food packaging were developed using PLA film with a weight of 5% Poly-Caprolaktone (PCL) employing thymol, carvacrol fillers, and their blend with supercritical CO_2_ fluids at 400 °C and 10 MPa for five hours. The PLA film is 5% by weight. The findings showed that due to the plasticizing impact of the compound induced by the supercritical CO_2_ fluid, the resulting film’s traction strength dropped. This supercritical CO_2_ fluid forms more porous microstructures and modifies its mechanical characteristics due to structural discontinuities which result in reduced flexibility and cracking resistance [15,16].

The antioxidant activity of the samples submerged in this coating type was greater. The improvement in physical-mechanical properties with the addition of filler must meet the following requirements: the filler must be miscible with PLA in order to form a homogeneous mixture; the filler must not be too volatile because it will cause evaporation when the temperature is raised (elevated temperature) during the process and the filler must not be easily migratory [17]. Natural extracts and other compounds, as previously stated, can be added to edible films, and the films can gain active properties, extending the shelf life of food through the migration of bioactive compounds or intelligent properties. This also enables the detection of food contamination by changing the color of the packaging [18].

Until now, there has been a lot of research into using PLA in food packaging as a biodegradable polymer to improve the mechanical properties of the film. The interaction of PLA and chitosan as antimicrobial agents against various target organisms such as algae, bacteria, yeast, and fungi in experiments involving in vivo and in vitro interactions with chitosan in various forms (solutions, films, and composites). TEO, because of their non-toxicity, biocompatibility, and high antibacterial and antioxidant capabilities, natural essential oils are also ideal candidates to replace frequently used food preservatives. The use of essential oils in active packaging applications is a cost-effective solution that can help to decrease food safety hazards. Furthermore, the use of natural essential oils in active packaging materials can help to preserve packed food from oxidation and protect against contamination and spoilage by microbes that cause food to perish and harmful free radicals. Antimicrobials have an effect on the properties of blocking materials from being exposed to bacteria in food packaging, thereby increasing shelf life and product quality. Therefore, the novelty of this study is in determining and examining the effect of the addition of turmeric essential oil mixed with chitosan on making food packagings using their “compatible” characteristics as a vitamin and a mineral. However, on the basis of literature investigations, no study has yet been carried out to change the physical and mechanical characteristics of environmentally friendly polymers in PLA and chitosan–TEO “green fillers.” The objective of the study was to investigate whether concentrations of turmeric extract were effective against *E. coli* and *Staphylococcus aureus*. *Staphylococcus aureus* is the most common cause of food poisoning globally. These bacteria can infect some foods, such as minimally processed ready-to-eat vegetables and processed meat products, and create a variety of enterotoxins. From the standpoint of public health, *Escherichia coli* as an intestinal pathogen is becoming increasingly significant, particularly the psychrotropic strain of *E. coli O157:H7* that can thrive on minimally processed vegetables and processed meat products.

## 2. Materials and Methods

### 2.1. Materials

Glycerol, curcumin, and gelatin were supplied by Sigma-Aldrich (Jakarta Timur, Indonesia), while turmeric, shrimp skin, 1% acetic acid, and commercial cassava flour were supplied by PT, Fugha Pratama Mandiri (Lhokseumawe, Indonesia). Phosphate-buffered saline (PBS), tryptic soybean broth (TSB), and tryptic soy agar (TSA) were purchased from Fisher Scientific (Jakarta, Indonesia). Medium molecular weight chitosan (molecular weight 190–310 kDa, 75–85% deacetylation), grafted polyethylene maleic anhydride (PEgMA) with 3.5% maleic anhydride content, and glycerol were supplied from Sigma Aldrich (Jakarta Timur, Indonesia). Furthermore, the crystalline PLA was supplied from NatureWorks Co. (Tangerang, Indonesia), while NaOH reagent was supplied from Sigma Aldrich (Jakarta Timur, Indonesia) without dilution.

### 2.2. Methods

#### 2.2.1. Chitosan Synthesis

An amount of 100 g of shrimp skin was cleaned with running water. Next, the shrimp skin was dried in an oven for 2 h at 160 °C, which was then smoothed. The demineralization process of shrimp skin powder was carried out using HCl concentrations of 0.25 M–2 M (ratio of 1:10 (b/v)) by heating and stirring for 1–2 h. After that, it was filtered and dried for 24 h. The deproteinization process uses 0.5 M–2 M NaOH solution with an immersion time of 10–400 min at temperatures of 20–100 °C. After the two procedures, the shrimp shell was filtered, washed with distilled water, and dried again to produce chitin powder. Then, the decolorization process was carried out using an acetone 1:10 (b/v) ratio and then immersed for 10 min. Chitin was dried in an oven for 2 h at 28 °C. The chitin powder obtained was sequentially bleached with 0.315% NaOCl for 5 min. Next, in the last deacetylation process, chitin powder was washed using 50% NaOH with a ratio of 1:20 (b/v). Then, it was heated for 3–5 h at 80–100 °C and washed with distilled water and 80% alcohol. Finally, chitin powder was then filtered. The chitin powder produced was analyzed using FTIR to determine the chitosan functional group [19].

#### 2.2.2. Extraction of Turmeric

A total of 20 g of turmeric was mashed, which was then placed into Soxhlet. The process of isolating turmeric oil was carried out by adding 200 mL ethanol for 8 h until no condensate drops again, and the process temperature was maintained at 78 °C. Thus, turmeric insulation is maintained with the distillation process until the oil was obtained.

#### 2.2.3. Biofilms Manufacture

The biofilm manufacturing process in [20] was a success.Three samples were formed from the composition of 2 g chitosan, 0 mL TEO, and 0 mL glycerol (Biofilm 1); 3 g chitosan, 0.3 mL TEO, and 0.5 mL glycerol (Biofilm 2); and 4 g chitosan, 0.3 mL TEO and 0.5 mL glycerol (Biofilm 3). The chitosan powder obtained was then dissolved in 30 mL of 1% acetic acid and 70 mL of distilled water, which was then mixed with 20 g PLA and glycerol according to the composition. The solution was homogenized and stirred at 70 °C for 60 min until the film solution was entirely homogeneous. In the final step, the film solution was added to TEO. Then, the film solution was homogenized for 30 min with a magnetic stirrer. The film solution was poured into a mold cleaned with 96% ethanol. It was then dried in the oven at 35 °C for 45 min. The remaining dried film was removed from the mold, which was ready for analysis.

#### 2.2.4. Surface Morphological Analysis

Scanning electron microscopy is a tool that can form shadows on the surface of broken microscopic specimens. The surface structure of the fibers was observed using a JEOL-T20 microscope at State Polytechnic of Lhokseumawe laboratory (Lhokseumawe, Indonesia). To form a conductive sample, it is necessary to coat the sample with a thin layer of gold. Scanning electron analysis was carried out at 5–20 kV. Tensile specimens tested were placed on a glass preparation part which was then placed on an optical lamp enlarged up to 1000 times. It was then photographed on each surface of the broken specimen or its fracture. The analysis of the treatment effect on the surface structure of the fiber was carried out using a microscope.

#### 2.2.5. Fourier Transform Infrared (FTIR)

The sample’s infrared spectroscopy was obtained on a KBr pallet (measurement method) using a Shimadzu FTIR Spectrophotometer from Chemical Engineering’s Laboratory at State Polytechnic of Lhokseumawe (Lhokseumawe, Indonesia). The spectrum was seen in the range of 500–4000 cm^−1^ with a resolution of 2 cm^−1^ with an empty KBr melting background.

#### 2.2.6. Thermogravimetric Analysis (TGA)

Thermogravimetric analysis (TGA) was used to determine the decomposition temperature of the sample, which is c using a Perkin–Elmer instrument from Chemical Engineering’s Laboratory at State Polytechnic of Lhokseumawe (Lhokseumawe, Indonesia) at a frequency of 1 Hz from −98 °C to 140 °C and a heating rate of 5 °C/min. Samples were then cut into small sizes. Finally, the position of the maximum tan value was used to determine the decomposition temperature of the sample.

#### 2.2.7. Anti-Microbial Test

A method was developed for determination of the antibiotic susceptibility of anaerobic bacteria by use of a single-disc diffusion technique and incorporation of the inoculum in pour plates from Chemical Engineering’s Laboratory at State Polytechnic of Lhokseumawe (Lhokseumawe, Indonesia). The method was standardized by the correlation of zone diameters with minimal inhibitory concentrations determined in the samples [21]. The antibacterial activity of biofilms was investigated using the agar media method in Petri dishes. Bacterial cultures grown in the mid-logarithmic phase were placed on agar media. *Escherichia coli* and *Staphylococcus aureus* were injected into agar media. After solidification of the agar coating, perfume solutions (15 mm in diameter) with different concentrations (5% and 10% by weight) were placed on the surface of the agar. The layers were incubated at 25 °C for 0 days to 9 days.

#### 2.2.8. Tensile Strength

The tensile strength of PLA–Chitosan–TEO was tested using a tensile strength tool according to the ASTM D 638–99 procedure from Chemical Engineering’s Laboratory at State Polytechnic of Lhokseumawe (Lhokseumawe, Indonesia).

#### 2.2.9. Statistical Analysis Using Statistical Statistical Package for Social Science (SPSS)

To find out the results of the analysis of whether the characteristics of the biofilms have a significant effect or not, further analysis was carried out using SPSS (Statistical Package for Social Science) version 22.0 using the one-way ANOVA method. If the data are normally distributed and homogeneous, a one-way ANOVA test was carried out with a 95% confidence level.

All data regarding the mechanical test of the sample are listed in Table 1, Table 2, Table 3 and Table 4. Table 1 presents tensile strength data for all samples using ANOVA method, Table 2 presents tensile strength data for all samples using ANOVA method for difference samples), Table 3 presents data % elongation for all samples using ANOVA method and Table 4 presents the data for % elongation for all samples using ANOVA method for difference samples.

Figure 1 depicts the procedures required to produce chitin powder from shrimp shell waste using a variety of ways of methods.

Figure 2 describes the steps that need to be taken to produce chitosan, the process of converting the acetyl group (NHCOCH_3_) in chitin to an amine group (NH_2_) in chitosan with the addition of NaOH.

Figure 3 is a soxhletation method carried out by installing a soxhletation device tools. A number of raw materials are put into the cladding and 96% ethanol is added as a solvent.

Figure 4 shows the manufacture of the biofilms is done by applying the principle of solution thermodynamics where the initial state of the solution is stable and then undergoes instability (addition of filler) in the phase change process (demixing), solidification (solidification) and phase transition so that at the stage it becomes solid after the addition of a high concentration polymer.

## 3. Results and Discussion

### 3.1. Scanning Electron Microscopic Analysis (SEM)

The scanning process of electron microscopy is based on the principle that an electron pistol produces an anode electron beam. In the direction of the sample, the magnet lens focuses on electrons. The whole sample is scanned by focused electron shafts through the scanner coil. The sample emits new electrons that are then received and sent back to the monitor where the particle is sampled.

Based on Figure 1, there were no chitosan particles in the analyses of Biofilm 1. The morphology of biofilms on the surface had space as no addition of glycerol as a binder had been found. Chitosan spreads evenly with Biofilm 2 particles and has a tight structure by adding TEO and glycerol. The difference between images (a) and (b) is shown in the image taken at 50× magnification. From the picture, it can be seen that the surface of the sample is hollow and cracked. The (a) sample surface is quite dominant with white spots and cracks, and in Figure 1b, the white spots are more clearly visible than in the previous image. These white spots are chitosan that has not been spread evenly and the mixing has not been uniform. A larger hollow in biofilm 3’s surface appears. This proves that the mixture of materials in this sample has a fairly good level of homogeneity. In the morphology of this second sample, there are also just small numbers of white spots, which are chitosan and TEO that have been evenly distributed [26,27]. Biofilm 3’s particles are evenly distributed on the surface but have a smaller film layer because the glycerol is lower than in biofilm 2. According to [28], insufficient compositions in chitosan could spread evenly on the PLA polymer matrix with plasticizers and oregano oil. They were able to form homogeneous bonds that affected the properties. According to the findings of this study, TEO has been shown to bind biopolymer elements in chitosan into good bonds and produce complex biofilm surfaces when added to biofilms. The interaction of the two biopolymer elements demonstrates an attraction capable of replacing inorganic filmmakers [29,30]. The pore structure is a site where cell growth or proliferation occurs. Among the three sample outcomes, the third sample (biofilm 3) with a total composition variation of 4 g chitosan, 0.3 mL TEO, and 0.5 mL glycerol is the best sample based on morphology with the greatest structure and pores with greater porosity.

### 3.2. Fourier Infrared Spectroscopy Analysis (FTIR)

The results of the analysis of the characteristics of turmeric essential oil from the extraction method by spectroscopy to obtain the value of the absorption results are used in Table 5. Qualitative analysis of different organic compounds can be confirmed from the characteristics of the vibration band appearing in the infrared spectrum region at a certain frequency influenced by certain functional groups. The transmittance percentage corresponding to the wave number is summed up in the total attenuated in the reflected IR spectrum, as shown in the figure above. There is an intense broad peak in the range of 2500–3200 cm^−1^, specifically at 2414.88 cm^−1^, corresponding to the polymer hydroxyl (OH) group. Another intense and branching peak in the range of 2256.71–2086.96 cm^−1^ corresponds to methyl OH stretching and C-H bending, i.e., most of the aromatic compound and alcohol components. Other studies have shown that there is a functional group of turmeric essential oil that will be produced will depend on the species of curcuma used [31].

The figure below shows a graph of the analysis of the PLA functional group. The graph shows the results of the FTIR test that from the optimum sample tensile strength in PLA, there exists N-H, C-H, O-H and C=O mentioned in Table 6. Where the NH group is present at the wave number 3336.85 cm^−1^ with a wavelength range of 3300–3400 cm^−1^, the OH group is found at the number 3630.03 cm^−1^ in the wave range of 3584–3700 cm^−1^ and the CH group at the number 3630.03 cm^−1^ in the wave range 3584–3700 cm^−1^. The C=O group is discovered at 1793.80 cm^−1^ with a wave range of 1540–1870 cm^−1^, while the following group is located at 3294.42 cm^−1^ with a wave range of 3267–3333 cm^−1^. This indicates that no new functional groupings, but only PLA pure characteristics, have been discovered in line with research by [32].

Turmeric essential oil interacts with the polymer matrix by forming intermolecular hydrogen bonds between their terminal hydroxyl group and the carbonyl groups of the ester moieties of both PLA and chitosan as mentioned in Table 7., in line with research [33] C-O stretching of alcohols and carboxylic acids, and N-H wagging of primary and secondary amines. These functional groups were predicted because chitosan is made up mostly of them. As evidenced by the appearance of a broad band at 3503 cm^−1^ corresponding to phenolic OH stretching vibrations and an increased intensity and shift of bands attributed to C–O bending vibrations as show in Figure 2. The bands ascribed to isoprenoids’ out-of-plane C-H wagging vibrations emerged at 2918 cm^−^^1^ and 2922 cm^−^^1^ when turmeric oil was added and mixed at a high speed to bind the matrix and filler.

### 3.3. Thermogravimetric Analysis (TGA)

We examined TGA as a tool to evaluate differences in mass loss patterns in order to quantify differences in biofilm development. To the best of our knowledge, this is the first time TGA has been used to characterize a biofilm in Indonesia. Biofilm thickness is generally measured using confocal microscopy, which is a very efficient approach but necessitates expensive equipment and extensive training to produce high-quality pictures. TGA is frequently used for material characterisation since it needs little training and sample preparation and produces findings quickly. This is the temperature range in which bacterial organic matter is most likely to breakdown and burn; therefore, the difference is ascribed to a bacterial biofilm [34]. The TGA results are consistent with our findings. An active biofilm in direct contact with the carbon foam surface should generate a current by creating a potential difference between the anode and the cathode, as seen.

The researchers discovered a mass drop in humidity from 50 °C to 150 °C and a mass decrease from 250 °C to 400 °C owing to thermal deterioration of the two components. This is the temperature range in which bacterial organic matter is most likely to break down and burn; therefore, the difference is ascribed to a bacterial biofilm. As a result, the temperature of the biofilm breakdown was about 285 °C. Figure 3 depicts biofilms degraded at 285.55 °C with a mass removal of 1.313 mg at 2 g chitosan, 0 mL TEO, and 0 mL glycerol. Biofilm 2, on the other hand, disintegrated at 274.02 °C with a mass removal of 1.54 mg. Furthermore, at 315.74 °C, samples with a weight removal of 1.74 mg were degraded using biofilm 3.

This research is in line with [35], in which weight loss and DTG (a derivative of TGA curves) of chitosan and Mandarin Essensial Oil–chitosan nanoparticles are presented. Chitosan went through two distinct stages. The first stage with green line (50–150 °C) was associated with free water loss. The second step used temperatures ranging from 250 °C to 400 °C with pink line for chitosan dehydration and breakdown. Material structural modifications, such as the addition of Mandarin Essensial Oil, Tween20, and chitosan cross-linked with sodium tripolyphosphate show by black line, resulted in a new stage blue line (150–300 °C) in the chitosan nanoparticles and Mandarin Essensial Oil–chitosan nanoparticles.

The comparison between samples (a) and (b) based on Figure 4 shows that the physical change in the biofilm tested may have been the most degraded in the biofilm. The initial golden color became black after being burned. The sample’s weight was also lost throughout the breakdown process. Each sample was examined, and it is known that the sample was broken down, owing to the combustion process in the TGA test equipment, resulting in the loss of carbon, water, or volatile components throughout the analysis method. Other combinations, such as chitosan, TEO, and glycerol, also had an effect on the sample’s heat resistance.

### 3.4. Anti-Microbial Test

TEO additives to chitosan biofilms have been studied to enhance the temperature of decomposition as one of the criteria for biological degradation. Furthermore, because of its antiseptic properties for food, biofilms with antimicrobial compounds have great food preservation abilities. As a result, in addition to preserving food, it also ensures the safety, freshness, and longer shelf life of these goods. This is due to the presence of antimicrobial action in natural substances and extracts utilized as additions in chitosan products. As a result, the usage of chitosan as a packaging material has several applications and is aimed at food safety.

It is critical to determine the antimicrobial activity, composition, structure, and functional groups of such extracts. Clove oil, thyme, cinnamon, rose salad, sage, and vanillin are the most active substances in the battle against germs. Several investigations have shown that they have antimycotic, non-toxic, and anti-parasitic effects. They can also be linked to the function of those chemicals in plants, as well as the antibacterial capabilities of essential oils and their constituents [16,36].

Figure 5 presents microbial development through the 12-day testing process that occurred in B1 (biofilm 1), B2 (biofilm 2), and B3 (biofilm 3). The results showed that the lowest colony growth rate of 61 colonies/g *S. aureus* was found in B1 with the composition of 2 g of chitosan, 0 mL of TEO, and 0 mL of glycerol, followed by B2 with 3 g of chitosan, 0.3 mL of TEO, and 0.5 mL of the glycerol growth rate of 77 colonies/g *E. coli*. B3 biofilm showed the best results, wherein 4 g of chitosan, 0.3 mL of TEO and 0.5 mL of glycerol have been mixed. The test results showed a microbial reduction with the addition of chitosan compounds, and the addition of essential oil from turmeric increased the anti-bacterial activity. *S. aureus* is a Gram-positive bacteria that can attach to glass, metal, and plastic as abiotic surfaces and host tissues as biotic surfaces. The attachment of *S. aureus* to surfaces depends on the surface components of the bacterial microbes recognizing the adhesive matrix molecules for host proteins. To prevent the attachment of *S. aureus* to a surface through the matrix, the surface must be coated with anti-adhesion agents such as arylrhodamins, calcium chelators, silver nanoparticles, and chitosan [37,38,39].

Bacteria adhere to surfaces in the form of a biofilm; therefore, one of the procedures that can prevent *S. aureus* pathogenesis is to prevent it from adhering to biotic and abiotic surfaces. Antibiofilm vaccinations are increasingly being used to inhibit *S. aureus* biofilm development [40]. Extracellular polysaccharides or cell wall-associated proteins identified in the biofilm matrix have the ability to stimulate an immune response that protects against *S. aureus* infection. Resistance to *S. aureus* can be acquired by active or passive vaccination with surface polysaccharides [41]. In contrast to *S. aureus*, *E. coli* bacteria live in the digestive system as part of the microbiota and may be a dangerous adversary. The presence of this bacterium can cause intestinal and extraintestinal infections in people and animals [42]. Based on the information gathered, *E. coli* has been added to the list of microorganisms of worldwide concern that cause the most prevalent illnesses in a variety of contexts, including communities, hospitals, and foodborne diseases [43]. These studies stated that combining chitosan and oregano oil as an essential oil could decrease water vapor permeability, puncture, and tensile strength. Still, they increased anti-bacterial and microbial properties due to the curcumin content and its essential oils, which inhibited the growth of causative bacteria, such as Bacillus sp, Shigella dysmetria, *S. aureus*, and *E. coli* [44,45]. Compared to the previously explained study, it can be concluded that the addition of TEO had shown an increased anti-bacterial activity, which was to inhibit microbial growth in the sample. That was because the composite material was directly contaminated with the air, containing various microbe types that can affect both physical and chemical ingredients.

### 3.5. Effect of Variations in PLA–Chitosan–TEO Concentration on Mechanical Characteristics of Food Packaging

Tensile strength testing of food packaging is an important factor that must be studied further to determine its application in the food industry, particularly for applications to vegetables and fruits. Mobility mechanical properties are critical because they are required to support in vitro culture and implantation processes [46]. Furthermore, testing the mechanical properties of packaging materials is critical for determining the homogeneity of a polymer mixture and the mixed materials used in the manufacturing process. In this study, the mechanical characteristics of the PLA–chitosan–TEO material were analyzed by testing the tensile strength (Tensile Strength) of the sample using the UTM (Universal Testing Machine) in the Tensile Test Laboratory of the Chemical Engineering Department, Lhokseumawe State Polytechnic. The dimensions of the specimens used follow the dimensions of the specimens in ASTM D 638–99. In this graph, the highest tensile strength value was obtained in sample biofilm 3. The composition of each biofilm 1 material is 13.03 Mpa. Meanwhile, biofilm 2 (3 g of chitosan, 0.3 mL TEO and 0.5 mL of glycerol) and 3 (4 g of chitosan, 0.3 of TEO and 0.5 mL of glycerol) have tensile strength values of 15.15 MPa and 17.67 Mpa, respectively.

Figure 6 based on the three samples, the third sample has a large tensile strength value, which is because the high concentration of chitosan causes the tensile strength of the analysis to increase. The increase in tensile strength was due to the reduced water content in the bioplastic because it had been absorbed by chitosan as an organic polysaccharide. Therefore, the molecular structure of bioplastics becomes denser and more homogeneous, which causes the tensile strength to increase [47,48]. This is also in line with conducted research which states that the addition of chitosan in PLA causes the tensile strength to increase based on research that has been carried out by [49] regarding the effect of the homogenization method and the content of carvacrol on the microstructure and physical properties of chitosan-based films. Film-forming emulsions were made with chitosan (1.5%), Tween 80 (0.5%), and carvacrol (0.25%, 0.5%, and 1.0%). The homogenization method used is the rotor-stator with high- and low-pressure homogenization. The results showed that the use of carvacrol had a significant impact on the mechanical properties of the emulsified films. The tensile strength produced by the film was significantly reduced. The results of adding chitosan to bioplastics showed that there was an interaction in the mixed film (starch–chitosan bioplastic). Chitosan has a good filming property and has extensive water absorption in the acidic medium due to the chain relaxation effect generated in the macromolecular networks by protonated amino groups. The addition of chitosan can increase the tensile strength of bioplastics [50]. However, if more chitosan is added, the more the tensile strength value of the bioplastics will decrease. In other words, the resulting bioplastic will have brittle properties [51]. The table below is in line with research conducted by [52]: the strength of the prepared samples was not affected by the type of natural extract, but only by the used basic matrix/polymer. The explanation of different textural properties can be affected by the different pH since the preparation of films included PLA.

The concentration of chitosan also affected the % elongation in PLA–chitosan–TEO biofilms samples. Elongation is a value that states the ability of a sample to be able to extend from its original shape or size as mentioned in Table 8. Figure 7 shows the elongation (%) of the PLA–chitosan–TEO biofilms obtained ranged from 18% to 25%. At a sample concentration of 2 gr chitosan, the lowest % extension value is obtained at a concentration of 2 g is only 18%. The chitosan concentration of 3 g, namely 22.7%, was the middle elongation obtained. The highest elongation came from biofilm 3 with 25% elongation. This is in line with the study of [53], where the minimum amount of chitosan used was set at 2 (by wt). It should be noted that starch could not be formed into films by glycerol solution. However, chitosan has an exceptional capacity to create film. As a result, it is also mentioned that biodegradable films were created using starch, and chitosan will provide a much better film because the films exhibited high strength and optimal elongation values. The results show that adding extracts to TEO and chitosan matrices improves the elongation and tensile strength of the biofilms formed. To decrease the resultant error value, each sample test was performed three times.

### 3.6. Techno Economic Challenges of the Developed and Future Research Directions

Biofilms are complex microbial communities composed of one or more species submerged in an extracellular matrix with varying compositions based on the kind of food-producing environment and the invading species. Bacteria and fungi are examples of microorganisms that may form these biofilms. The presence of many bacterial species in a biofilm has significant ecological benefits since it facilitates the biofilm’s adhesion to a surface. This can also happen in the absence of specific fimbriae in some species. Disinfectants such as quaternary ammonium compounds and other biocides are more resistant to mixed biofilms. In the food business, biofilms may build swiftly. The first two phases condition the material’s surface and the reversible cell binding to that surface. The binding then becomes permanent, and the formation of microcolonies begins. Finally, the tridimensional structure of the biofilm is established, giving rise to a sophisticated ecosystem suitable for dispersal. Some biofilm-forming bacteria in food factory settings cause human diseases, which is very important in the food business. These pathogens may form biofilm structures on a variety of artificial substrates used in the food business, including stainless steel, polyethylene, wood, glass, and polypropylene.

Prior to selection, the in situ relevance of the biofilm model must also be reviewed. For example, before employing a specific model, it should be evaluated if the inoculum, flow regime (if any), and surfaces/interfaces fit the environment in the system being represented. To achieve in situ relevance, the essential drivers and factors in the system and environment must be recognized. Tools for evaluating how the model and results may be translated to practice should be established, and practitioners (including doctors and engineers) should be included in model review to guarantee in situ relevance. With an ever-increasing number of biofilm-related articles being published and a wide range of various methodologies becoming accessible, it can be difficult for researchers new to the subject (or to a specific subdiscipline within the field) to select the best appropriate model system. In addition, some instruction on data interpretation may be required (e.g., how much biofilm reduction is needed in a particular model before considering it biologically meaningful). Although biofilms have an influence on many elements of life and biological sciences, they are considered a specific discipline.

### 3.7. Comparative Table of Materials Used for Food Packaging Applications

Table 9 is represent the difference process of producing food packaging with essential oil fillers as anti-bacterial and anti-oxidant chemicals that can extend the shelf life of components and protect product quality from free radicals.

### 3.8. Comparison between Produced Materials and Commercials Product

The FTIR study of chitosan yielded the absorption regions of functional groups, as shown in Figure 8. The the red line shows that stretching OH group absorption emerged at a wave number of 3189.37 cm^−1^ with a deacetylation temperature treatment of 100 °C for two hours, and the hydroxyl group (-OH) appeared at a wave number of 2980.20 cm^−1^. The N-H stretching group may be found at wave number 3004.61 cm^−1^. The vibration of the C-H range on aliphatic CH_2_ is shown by the absorption at a wave number of 2779.37 cm^−1^. The appearance of bending vibration absorption CH_2_ at a wave number of 1365.75 cm^−1^ supports this. The absorption at 1567.09 cm^−1^ indicates the C=O group (amide peak) that remains because the chitosan generated has not been entirely deacetylated. In the isolated chitosan IR spectra, N-H bending vibration of NH_2_ is represented by the 1376 cm^−1^. The bending absorption of -CH_3_ is modest and visible at the wave number 1389.45 cm^−1^. At the wave number 1306.09 cm^−1^, C-N stretching vibrations with low intensity were detected. The C-O bond range was discovered at wave numbers 1155.36 cm^−1^ and 1114.86 cm^−1^.

Chitosan at a deacetylation temperature of 100 °C from commercial chitosan for two hours (black line) showed absorption of the stretching OH group at a wave number of 2992.11 cm^−^^1^ and a hydroxyl group (-OH) at a wave number of 3180.91 cm^−^^1^. The N-H stretching group emerges at the wave number 3003.43 cm^−^^1^. Absorption at wavelength numbers 2945.87 cm^−^^1^ and 2790. cm^−^^1^ demonstrates the C-H stretching vibration on aliphatic CH_2_, which is reinforced by the presence of bending vibration absorption CH2 at wave number 1398.44 cm^−^^1^. The absorption at the wave number 1590.34 cm^−^^1^ indicates the C=O group (amide peak) that remains because the chitosan generated has not been entirely deacetylated.

The area for pure substances is greater than for commercial products. The Figure 9 shows that the yield of the resultant extract increases, as well as the number of extraction steps, with the extraction stage, yielding more extract than the 1-stage extraction. Extraction with fewer solvents will be more effective than extraction with all the solvents at once. This is due to the fact that at each stage, there will be contact with a new solvent, which provides a driving force in the form of variations in concentration and solubility at each stage, ensuring that a solute is always transferred from solid to solvent. The extract yield will eventually fall like Figure 10. This is due to the fact that in crosscurrent, multistage extraction, the solids employed at each stage are the same solids; therefore, the longer the extract is extracted, the more saturated it becomes until the extract gain no longer grows and decreases.

## 4. Conclusions

Based on the research results on PLA as a biofilm, it was concluded that the maximum decomposition temperature obtained was at the temperature of 315.74 °C with variations of 4 g chitosan, 0.3 mL TEO, and 0.5 mL glycerol (biofilm 3). Molecules produce solid particles and are evenly distributed around the surface, affecting anti-microbes’ properties. Based on the anti-microbial test, chitosan and TEO showed an increase in anti-bacterial activity. Moreover, the biofilms produced were able to fight *S. aureus* and *E. coli* bacteria in 9 days of exposure to open spaces. The concentration of chitosan affects the tensile strength of the PLA–chitosan–TEO sample: a higher concentration value of chitosan will make the bioscaffold material have an increased tensile strength value. The highest tensile strength value was in biofilm 3, with a composition of 4 g of chitosan concentration. However, if the addition of chitosan is continuous, it will increase the brittleness through higher water absorption so that further studies on the maximum amounts of chitosan as a filler in the manufacture of food packaging need to be considered. Biofilms in the food business are a severe economic and health concern. On the one hand, the presence of biofilms of food processing surfaces might result in financial losses due to corrosion on metal surfaces caused by certain bacteria, necessitating the replacement of these parts. Furthermore, certain bacterial species, such as Pseudomonas spp. and Bacillus spp., express a variety of proteolytic and lipolytic enzymes that can produce unpleasant smells and tastes (rancid, bitter). The impacted production batches must be removed and destroyed in these cases. On the other hand, and more importantly, biofilm development in food manufacturing is a critical public health issue. These biofilms may comprise bacterial (and occasionally fungal) species known to be harmful in healthy people, or they may solely target the immunocompromised (such as organ transplant recipients, oncology or HIV patients, etc.). These bacteria can cause food poisoning *(S. aureus)* as well as gastroenteritis *(E. coli)* and systemic illnesses in some situations *(E. coli* O157:H7).

## Data Availability

The data presented in this study are available on request from the corresponding author.

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
