# Peer review of "Biodegradation of Polylactic Acid-Based Bio Composites Reinforced with Chitosan and Essential Oils as Anti-Microbial Material for Food Packaging"

_polymers, 2021, doi:10.3390/polym13224019_

Round 1
Reviewer 1 Report
I read an interesting research work entitled Biodegradation of PLA Based Bio Composites Reinforced with Chitosan and Essential Oils as Anti-Microbial Material for Food Packaging. The concept of the article is interesting and suitable to publish in Polymers. This manuscript is generally not well written and needs to address following comments, and thus require substantial major revision of the manuscript.
- Abstract should be rewrite adding more detail of results and importance of the study.
- A well addressed graphical scheme of study design should be inserted.
- Add more keywords related to the research work
- In the introduction section, write the novelty of the work and the problem statement clearly. More discussion about the synthetic plastics pollution, add details about bioplastics particularly PHA and their importance in food packaging material for this author can refer and cite recent review article Bioresource Technology Volume 325, April 2021, 124685. Give detailed research objectives at the end of introduction not the repetition of abstract.
- Line no 93 why authors cited reference no 42? 4th para of introduction looks like the authors discussed their result delete the same. Give details of chemicals, instruments as well as reagents throughout the manuscript.
- Why authors extracted the chitosan and TEO are they in pure form. In this case if they will compare their results with pure chemical would be better
- Table 1 is not correct. SEM image presentation is very poor need to give details in the figure
- Substantial discussion of FTIR peaks denote the values in figure and their comparison with the literature is expected during revision. Overlap all ftir in one image
- Figure 4 what the TGA means? It's very confusing
- Have authors checked the stability of produced materials?
- Add one Comparative table of materials used for food packaging applications with your research outputs.
- Techno Economic challenges of the developed system need to be addressed. What are the limitations and future research directions that need to be described by adding a new section before the conclusions section?
- The conclusion of the study needs to be added with the specific output obtained from the study, it could be modified with precise outcomes with a take home message.
- Some English and grammar mistakes are present that need to be correct to improve the quality of the manuscript.
Reviewer 2 Report
Manuscript deals with an interesting topic. There is also clearly visible application potential. However, several points have to be correcter prior consideration to publish the above mentioned manuscript:
- part 2.1: Used materials are of unknown source. For every reagent, supplier, city and country of origin has to be given. The information about purity or purification would be helpful.
- part 2.2: The same for instruments. There has to be clearly described type, manufacturer, city and country of the instrument origin. For US or Canada manufacturers also state.
- The tables 1 and 2 have to be copletely rewritten to some acceptable format. As it is, every column is of different style and therefore any orientation is very hard.
- Graphs: Presented graphs have to be unified as it is possible. Each graph is of different format, what is not acceptable.
- line 131 and 135 and others: there has to be written "°C". All forms which are used in manuscript are wrong. Please copy exactly this form everywhere.
All in all, the topic is interesting but the manuscript is not well written and several points have to be changed to improve its overall quality.
Round 2
Reviewer 1 Report
Authors have answered my raised comments however for a few comments more clarification and proper explanation is highly expected.
1) Authors should mention in the answer whatever the changes they have made with page no and line no
2) It looks very surprising if the reviewer asks to modify the abstract then the author highlights the whole section which is not good. Highlight the text which has been modified.
3) It is worth it to extract the biomolecules however my question is about the purity of the extracted biomolecules so it's better to compare the results with commercial biomolecules.
4) Give proper answer for Have authors checked the stability of produced materials?
5) for this comment why the author adds only one literature which is quite surprising add more examples and give research highlights not the abstract.
6) EM image presentation is very poor need to give details in the figure itself
7) Still Some English and grammar mistakes are present that need to be correct to improve the quality of the manuscript
Reviewer 2 Report
The manuscript is much better now and can be considered for publication.
Author Response
Thanks for your consideration and your hard work for our manuscript
Round 3
Reviewer 1 Report
The authors have substantially revised the manuscript according to the comments.
The present form of the manuscript can be accepted for publication.
This manuscript is a resubmission of an earlier submission. The following is a list of the peer review reports and author responses from that submission.
Round 1
Reviewer 1 Report
- The abstract section is poorly constructed. No clear information about the studied materials.
- The authors should give more precise details about the purpose of the work in the introduction section. The novelty or significance of this work is not clearly stated. More recent literature needs to be updated.
- Figure 1- SEM image- higher magnifications images required for studied materials.
- Many space errors/punctuation errors must be solved. The abbreviations should be checked in the manuscript and make clear. SEM defined several times.
- Figure resolution need to be improved. All the figure captions must be discussed in more detail with experimental conditions. Figure 2,3- hard to differentiate the measured plots. Figure 5- its not uniform with earlier figures.
- Mechanical properties of the materials need to be discussed.
- The Authors are encouraged to review the form and the manuscript's English.
- During the presentation of the results, please provide a comparison with previous similar materials. Authors have to compare the latest literature. Why should these biocomposite materials be better than the others?
- Conclusion section- must focus on future directions of the studied materials?
Author Response
- we are so sorry, but we just edit the abstract and the studied materials theory has been put inside.
- we just edit the whole introduction part including the novelty of the research.
- we are so sorry to inform you that that image is real based on the SEM test.
- All error has been edit include SEM abbreviations
- figure caption has been edited and the size of a figure has been adjusted
- we just discussed the Mechanical Properties of the materials
- The review already implemented
- The study has been comparing the latest literature.
- The conclusion has been improved
Reviewer 2 Report
My comments are the following:
- Additional information about chitosan edible/biodegradable films should be written. The following reference should be used: Jancikova, S., Dordevic, D., Tesikova, K., Antonic, B., & Tremlova, B. (2021). Active Edible Films Fortified with Natural Extracts: Case Study with Fresh-Cut Apple Pieces. Membranes, 11(9), 684.
- Line 134: how films were dried?
- Line 164: which colonies?
- There is no statistical analysis, it must be included in the work.
- Mean values and standard deviations are not present in the manuscript too.
- Discussion part is very short, same as number of references is only 23.
Author Response
- The reference has been cited in the manuscript
- The film is then dried in the oven at 350C for 45 minutes
- The colonies is S.Aureus and E.Coli
- We collect
- We just add the mechanical properties of the sample so there is a comparison between our study and the latest study
- Mean values and standard deviations already added
- The references already improved
Round 2
Reviewer 2 Report
The manuscript cannot be accepted for the publication.